# Nutritional Status Predicts Fatty Acid Uptake from Fish and Soybean Oil Supplements for Treatment of Cancer-Related Fatigue: Results from a Phase II Nationwide Study

**DOI:** 10.3390/nu14010184

**Published:** 2021-12-31

**Authors:** Amber S. Kleckner, Eva Culakova, Ian R. Kleckner, Elizabeth K. Belcher, Wendy Demark-Wahnefried, Elizabeth A. Parker, Gilbert D. A. Padula, Mary Ontko, Michelle C. Janelsins, Karen M. Mustian, Luke J. Peppone

**Affiliations:** 1Department of Pain and Translational Symptom Science, University of Maryland School of Nursing, Baltimore, MD 21201, USA; ian.kleckner@umaryland.edu; 2Department of Surgery, Division of Supportive Care in Cancer, University of Rochester Medical Center, Rochester, NY 14642, USA; eva_culakova@urmc.rochester.edu (E.C.); michelle_janelsins@urmc.rochester.edu (M.C.J.); karen_mustian@urmc.rochester.edu (K.M.M.); luke_peppone@urmc.rochester.edu (L.J.P.); 3Department of Psychological Science, Hobart and William Smith Colleges, Geneva, NY 14456, USA; belcher@hws.edu; 4Department of Nutrition Sciences, University of Alabama at Birmingham, Birmingham, AL 35233, USA; demark@uab.edu; 5Department of Physical Therapy and Rehabilitation Science, School of Medicine, University of Maryland, Baltimore, Baltimore, MD 21201, USA; elizabeth.parker@som.umaryland.edu; 6Department of Radiation Oncology, University Hospitals St. John Medical Center, Westlake, OH 44145, USA; gilpadulamd@gmail.com; 7Dayton Clinical Oncology Program, Dayton, OH 45420, USA; mary.ontko@daytonncorp.org

**Keywords:** breast cancer, survivorship, fish oil, omega-3, nutritional status, malnutrition

## Abstract

Cancer-related fatigue is a prevalent and debilitating condition that persists for years into survivorship. Studies evaluating both fish oil supplementation on fatigue and associations between fish oil consumption and fatigue have shown mixed effects; it is unknown what factors contribute to these differential effects. Herein, we investigate whether the nutritional status of cancer survivors was associated with serum omega-3 concentration or change in serum omega-3s throughout a fish oil supplementation study, and then if any of these factors were associated with fatigue. Breast cancer survivors 4–36 months post-treatment with moderate-severe fatigue were randomized to take 6 g fish oil, 6 g soybean oil, or 3 g of each daily for 6 weeks. Baseline nutritional status was calculated using the Controlling Nutritional Status tool (serum albumin, lymphocytes, cholesterol). At baseline and post-intervention, serum fatty acids were quantified and fatigue was assessed using the Multidimensional Fatigue Symptom Inventory. Participants (*n* = 85) were 61.2 ± 9.7 years old with a body mass index of 31.9 ± 6.7 kg/m^2^; 69% had a good nutritional score and 31% had light-moderate malnutrition. Those with good nutritional status had greater total serum omega-3s at baseline (*p* = 0.013) and a greater increase in serum omega-3s with supplementation (*p* = 0.003). Among those who were supplemented with fish oil, greater increases in serum omega-3s were associated with greater improvements in fatigue. In conclusion, good nutritional status may increase uptake of fatty acid supplements, increasing their ability to improve fatigue.

## 1. Introduction

Cancer-related fatigue affects approximately half of patients with cancer, with greater prevalence in those who undergo chemotherapy and/or radiation treatment [1,2,3]. For breast cancer, specifically, fatigue is one of the most prevalent and distressing side effects of the cancer experience, and often persists for years into survivorship [4]. By definition, cancer-related fatigue cannot be relieved by sleep or rest, it can impair the ability to perform activities of daily living, and it can reduce quality of life [5,6]. There is a lack of knowledge regarding the etiology and pathophysiology of cancer-related fatigue, thereby preventing the development of effective preventive strategies and treatments [7].

Recently, it has been suggested that nutritional interventions such as a diet high in fruits, vegetables, nuts, seeds, and fish may improve cancer-related fatigue [8,9]. Indeed, malnutrition—specifically low recent protein intake [10] and low plasma glutamine [11]—is implicated in the development of cancer-related fatigue. However, details regarding what specific nutrients are beneficial and who will benefit from these interventions are still lacking. Specifically, there are mixed data regarding whether omega-3 fatty acid supplementation can help improve cancer-related fatigue. In two cross-sectional studies, higher intake of omega-3 fatty acids or a higher omega-3/omega-6 ratio, as estimated with a food and supplement frequency questionnaire, was associated with less fatigue [12,13]. Similarly, a 3-month dietary intervention for breast cancer survivors that included omega-3 fatty acid-rich foods, among other ingredients, resulted in a reduction in fatigue compared to a general health education control arm [8]. Our group performed a phase II multisite three-arm randomized controlled trial comparing 6 g fish oil, 3 g fish oil + 3 g soybean oil, or 6 g soybean oil supplementation per day for 6 weeks among breast cancer survivors; this was the largest clinical trial among breast cancer survivors to date that tested fish oil as an experimental intervention [14]. Fatigue improved for participants in all three groups, but there were greater improvements in fatigue with soybean oil compared to fish oil [14]. Thus, there are mixed results on the use of omega-3 fatty acids in the treatment of fatigue, and it is unknown what factors contribute to these differential effects.

We hypothesize that nutritional status influences omega-3 supplement uptake and metabolism, thereby affecting the ability of omega-3 supplementation to reduce fatigue. Biochemical indicators of nutritional status are useful and commonly used in clinical oncology practices. Specifically, the Controlling for Nutritional Status (CONUT) score is a common screening tool for malnutrition [15]. It is comprised of measurements of serum albumin, lymphocytes, and cholesterol, and has been associated with poor prognostic outcomes including poor surgical outcomes in patients with cancer [15,16,17,18]. The combination of these measures is useful because it integrates three disparate yet related measures of the availability of resources. Albumin is a common indicator of protein reserves, though it is also affected by other diverse pathologies [19,20]. A low total lymphocyte count can reflect inadequate biological resources to launch an appropriate immune response [21,22]. Total cholesterol is positively correlated with body mass index (BMI) and low circulating cholesterol is associated with recent weight loss; therefore lower total cholesterol can be considered an indicator of low energy intake and caloric reserves [23].

The goal of this study was to investigate whether nutritional status was associated with uptake of the fatty acids or the associations between serum omega-3 concentration and fatigue. We hypothesize that good nutritional status is associated with greater increases in serum omega-3 fatty acids over 6 weeks of fish oil supplementation and greater improvements in fatigue. Thus, we conducted a secondary analysis of data from our existing randomized omega-3 supplementation study among female breast cancer survivors to begin to test our hypotheses.

## 2. Materials and Methods

### 2.1. Study Design and Population

This was a secondary analysis of a phase II multisite randomized controlled trial (NCT02352779). This study was conducted through the University of Rochester Cancer Center (URCC) National Cancer Institute (NCI) Community Oncology Research Program (NCORP) Research Base to assess the effects of fish vs. soybean oil on persistent cancer-related fatigue. The full methods and primary aims of the parent study have been published previously [14]. Briefly, female breast cancer survivors were recruited who were 18 years old or older, had had stage 0-III cancer, were 4–36 months post-treatment (surgery, radiation, and/or chemotherapy), had self-reported fatigue ≥ 4 on a visual analog scale from 0–10, and had not taken omega-3 supplements in the previous 12 weeks. Participants were allocated 1:1:1 to one of three supplement groups: 6 g fish oil, 6 g soybean oil (control), or 3 g of each daily for 6 weeks. Randomization was accomplished by random algorithm and participants were stratified by baseline fatigue level (two levels: 4–6 (moderate) or ≥7 (severe) on a 11-point visual analog scale anchored by 0 (no fatigue) and 10 (worst possible fatigue)). Participants were included in this analysis if data were available for serum fatty acids, albumin, lymphocytes, and cholesterol at baseline (available for 85/108, 78.7%).

### 2.2. Measures

Blood samples were collected after an overnight fast at baseline and post-intervention (6 weeks). Omega-3 fatty acids, omega-6 fatty acids, and cholesterol were quantified via the Serum Comprehensive Fatty Acid panel by Mayo Clinic Laboratories (Rochester, MN, USA) as described previously [14]. Albumin, total lymphocytes, height, and weight were measured as part of participants’ routine medical treatment and results were extracted from their medical record at baseline, which had been updated within the last 3 months (average <1 month before baseline). Baseline nutritional status was calculated using the Controlling for Nutritional (CONUT) Status tool [15]: scores were assigned in the categories of serum albumin, total lymphocytes, and cholesterol, where higher scores indicate lower concentrations. As per standard cut-offs, a score of 0–1 indicated good nutritional status, 2–4 indicated light malnutrition, 5–8 indicated moderate malnutrition, and 9–12 indicated severe malnutrition [15].

The Multidimensional Fatigue Symptom Inventory-Short Form (MFSI) [24,25] was used to assess fatigue at baseline and post-intervention. The MFSI is an empirically derived 30-item questionnaire that assesses five subdomains—general, physical, mental, and emotional fatigue as well as vigor. Items are rated on a five-point scale from 0, “Not at all,” to 4, “Very much,” and then the items are scored for each subdomain. The questionnaire yields a total score that ranges from −24 to 96 with a greater score indicating greater fatigue. The MFSI has demonstrated validity and reliability among patients with cancer [24,25].

### 2.3. Statistical Analysis

The distribution of baseline characteristics was evaluated for those with good nutritional status vs. light-moderate malnutritional status (there were no participants with severe malnutrition); the mean ± standard deviation (SD) and *n* and percent of total *n* are reported for continuous and categorical measures, respectively. The difference in changes in fatty acids by arm (0, 3, or 6 g fish oil) were compared using analysis of variance (ANOVA). Regression modeling was used to assess the association between baseline nutritional status and fatty acid uptake; the models were adjusted for study arm (Change in fatty acid = CONUT + Fish Oil Dose). To account for outliers among the fatty acid measures, the robust M-estimation method with Huber weight function was used [26]. Robust regression with M-estimation was also used to evaluate associations between changes in fatty acids and changes in fatigue, adjusting for age and BMI. Lastly, ANOVA was used to compare the change in total fatigue and subscales among those with good vs. light-moderate malnutrition. All analyses were performed using SAS (version 9.4, SAS Institute, Cary, NC, USA) and JMP Pro (version 14.1.0, SAS Institute). For this analysis, a *p*-value of <0.05 was deemed statistically significant. Due to the exploratory nature of this analysis, small sample size, and focus on biological relationships, no multiplicity adjustments were performed.

## 3. Results

### 3.1. Demographics and Clinical Characteristics

Participants (*n* = 85) were 61.2 ± 9.7 years old and had a BMI of 31.9 ± 6.7 kg/m^2^. Approximately two-thirds (*n* = 59) had good nutrition status and 31% (*n* = 26) had a light or moderate malnutrition score (Table 1 and Figure 1). A higher percentage of participants on hormonal therapy had light-moderate malnutrition than those not currently on this treatment (36.5% vs. 13.6%, χ^2^ = 0.045). Notably, BMI was not associated with nutrition status (31.5 ± 6.0 kg/m^2^ for those with good nutrition status and 32.8 ± 8.3 kg/m^2^ for those for light-moderate nutritional status, *p* = 0.47).

At baseline, those with good nutrition status had a greater concentration of circulating omega-3 fatty acids (mean ± SD = 0.33 ± 0.10 mM vs. 0.27 ± 0.12 mM, *p* = 0.013, Table 1) with markedly greater concentrations of EPA (70.7 ± 34.1 µM vs. 48.1 ± 35.2 µM, *p* = 0.009). Those with good nutrition status also had greater concentrations of the common dietary omega-6 fatty acid linoleic acid (3.50 ± 0.56 mM vs. 2.96 ± 0.55 mM, *p* < 0.001) as well as the omega-6 fatty acid arachidonic acid (845 ± 243 µM vs. 721 ± 241 µM, *p* = 0.035). The ratio of omega-6:omega-3 ratio was similar between those with good nutrition status vs. light-moderate malnutrition (14.9 ± 4.1 vs. 16.7 ± 6.1, *p* = 0.21).

### 3.2. Change in Circulating Fatty Acids with Supplementation

Supplementation with 3 g or 6 g fish oil daily effectively increased total serum omega-3 fatty acids (including DHA and EPA), while 6 g soybean oil daily increased total omega-6 fatty acids (*p* < 0.01, Table 2). Fish oil supplementation (3 g or 6 g daily) also significantly decreased the omega-6:omega-3 (*p* < 0.001) and arachidonic acid (*p* = 0.001) in a dose-dependent manner. Importantly, better nutritional status was associated with a greater increase in omega-3s with supplementation (total omega-3s, *p* = 0.005; DHA, *p* = 0.003; and EPA, *p* = 0.032; Table 3, Figure 2). However, nutritional status was not associated with changes in total omega-6 concentrations (total, linoleic, or arachidonic; *p* > 0.32) or the omega-6:omega-3 ratio (*p* = 0.21, Table 3). Our model estimates that breast cancer patients with good nutrition status (CONUT = 0) vs. those with malnutrition (CONUT = 3) who supplement with 3 g of fish oil daily for 6 weeks would see an increase of 0.36 mM vs. only 0.21 mg/mL circulating omega-3 fatty acids—a 70% greater increase. Extrapolating these models, a breast cancer survivor with a CONUT score of 3 would theoretically have to take 64% more fish oil to get the same 0.3-mM increase in circulating omega-3′s than a survivor with good nutrition status (a CONUT score of 0).

### 3.3. Associations between Nutritional Status, Omega-3 Uptake, and Cancer-Related Fatigue

Fatigue, as measured using the MFSI, significantly improved over time for participants in all groups (p < 0.001) and the effect of the group was not significant (*p* > 0.50), as we previously reported [14]. Nutritional status at baseline was not associated with improvements in fatigue for MFSI total score or any subscales (Appendix A). Notably, for those who supplemented with fish oil, greater changes in omega-3 fatty acids in the blood were associated with greater improvements in fatigue over the course of the intervention, especially greater increases in physical fatigue (β ± SE = 4.53 ± 1.98, *p* = 0.022) and vigor (β ± SE = 4.85 ± 2.04, *p* = 0.018, Table 4).

## 4. Discussion

This analysis, which was conducted on the largest fish oil supplementation trial among breast cancer survivors to date, suggests that survivors with better nutritional status had significantly greater increases in total serum omega-3 fatty acids from fish oil supplements. Among those who were supplemented with fish oil, greater increases in serum omega-3 fatty acids were associated with greater improvements in fatigue. This study supports the hypothesis that nutritional status affects the uptake of bioactive omega-3 fatty acids and modulates their effects on cancer-related fatigue. These results highlight that nutritional support should not narrowly focus on single supplements, but should more broadly consider how we can increase the ability of patients to absorb and utilize nutrients that we believe are beneficial.

Malnutrition is a prevalent yet underdiagnosed and undertreated condition in oncology [27], and it has a bidirectional relationship with nutrient malabsorption. Malnutrition is often difficult to recognize because it can be masked by excess body fat and can occur in the absence of weight loss [27,28]. Herein, 75/85 of our participants had a BMI greater than 25 kg/m^2^, which categorizes them as overweight or obese. However, 31% had light-moderate malnutrition, and BMI was not associated with malnutrition status (Table 1). These results reiterate the need for oncologists to continue to screen for malnutrition into survivorship using validated tools such as CONUT, phase angle, Nutritional Risk Screening 2002, the Malnutrition Screening Tool, or others [15,29]. Diagnosis of malnutrition will allow clinicians to treat it, improve health outcomes, and perhaps improve responsiveness to beneficial nutritional supplements. Deficiencies in nutrients such as iron, carnitine, and vitamin B12 are particularly common in patients undergoing cancer treatment [30]; these deficiencies can persist into survivorship and reduce quality of life. Early screening for nutrient deficiencies is key to diagnosis and early reversal.

Several other studies have demonstrated large variations in the uptake and utilization of omega-3 fatty acids from supplements and explored factors that predict uptake of omega-3′s from supplements. For example, a higher BMI is associated with lower increases in omega-3 concentrations from supplementation due to a greater volume of distribution of the fatty acid supplements [31]. Additionally, several studies in healthy adults have demonstrated that a lower omega-3 fatty acid concentration in plasma and/or erythrocyte membranes was associated with greater omega-3 fatty acid uptake from supplements [32,33,34]. In addition, older age, female sex, and more physical activity were associated with greater increases in omega-3 uptake from an EPA+DHA supplement [32]. We observed an association between active hormonal therapy and malnutrition as measured with CONUT (Table 1). Indeed, antiestrogens such as tamoxifen have been associated with reductions in serum insulin-like growth factor-1 (IGF-1), and low IGF-1 is associated with malnutrition [35]. In addition to demographics, behaviors, and clinical characteristics of the individual, the form of omega-3s and the food matrix (e.g., fatty fish in the diet, supplements, chemical structure) can affect bioavailability [36]. For example, dietary fat stimulates pancreatic and gall bladder secretions and promotes fat absorption; therefore omega-3s from fatty fish and fatty acid supplements consumed with a fatty meal are better absorbed than supplements taken with a low-fat meal [36]. Clinicians should consider patients’ baseline measures and clinical characteristics when recommending and dosing high- omega-3 supplements, as well as consider a referral to a dietitian to maximize nutritional status and effectiveness of their dietary supplements.

There is a substantial and growing body of literature that nutritional status modulates fatigue in the cancer population and more broadly [9]. For example, Azzolino et al. describe how exhaustion of metabolic reserves, either from macro- or micronutrient deficiencies, may be experienced as fatigue [37]. In a study among patients with colorectal cancer, laboratory markers of nutritional status were strongly associated with cancer-related fatigue [38]. Herein, among those who were supplemented with fish oil, we observed that greater increases in omega-3 fatty acids were associated with greater improvements in fatigue (Table 4). Similarly, as part of the Health, Eating, Activity, and Lifestyle (HEAL) study, a cross-sectional analysis of 633 breast cancer survivors revealed an association between higher intake of omega-3 fatty acids and lower odds of fatigue [12]. Also, in a randomized controlled trial that tested a “Fatigue-reduction diet,” which included high amounts of omega-3 fatty acids, fatigue improved among breast cancer survivors [8].

Improving uptake and utilization of fish oil supplements is only one benefit of improving nutritional status of cancer survivors. There is emerging evidence that healthy nutritional status is associated with improved cancer treatment outcomes, less severe symptoms, and higher quality of life [39,40]; therefore, by prioritizing the identification and treatment of malnutrition [27,28], the impact of a broad array of therapeutics ranging from lifestyle factors to pharmacologic agents could potentially be enhanced. Moreover, dietary interventions that improve both nutritional status and fatty acid profiles have wide-reaching health benefits with little to no side effects. A Mediterranean Diet, specifically, which is high in fish among other foods high in nutrient density, can improve fatty acid profile [41,42].

In addition to screening and addressing malnutrition, clinicians should screen for fatigue long into cancer survivorship. While multidimensional scales are the most useful for clinical research, a single question (e.g., *What was your average fatigue level in the past week on a scale from 0–10?*) can reveal a problem with persistent fatigue and open conversation regarding its potential cause(s) and solutions [43].

This analysis has several limitations that should be considered when interpreting our results. First, malnutrition can be assessed using various criteria, and CONUT does not capture body composition, recent weight loss, or other clinical measures [44]; indeed both obesity [45] and sarcopenia [46] contribute to fatigue and these features of malnutrition should not be ignored. Similarly, we did not assess incorporation of omega-3 fatty acids into plasma membranes (e.g., membrane content of erythrocytes), only circulating fatty acids. However, these measures are correlated [47]. Additionally, we do not have albumin or lymphocyte values, and therefore CONUT values, from our post-intervention time point, only baseline. Therefore, we could not evaluate how nutrition status changed over the 6-week intervention. However, baseline measures are very useful clinically. Next, this cohort was exclusively female and predominately White, non-smoking, and highly educated, so our data should not be generalized to other populations without prudence. With that said, our participants were recruited from multiple community sites across the United States, not just academic medical centers, as is the case with most clinical trials. Lastly, this was a secondary analysis that was exploratory and hypothesis-generating by nature; we acknowledge that we performed multiple comparisons and there is risk of increase type 1 error. Additionally, randomization did not occur by nutritional status. Therefore, our observed relationships between nutritional status, omega-3 uptake, and cancer-related fatigue should be tested for replication in future, hypothesis-testing studies.

Despite these limitations, this study has many unique strengths. This was the largest fish oil study among breast cancer survivors to date [14]; it has a strong randomized controlled design and includes a large sample of breast cancer survivors from community oncology clinics throughout the United States, which allows for generalizability of the results. Our primary outcome, the MFSI, is the gold standard for fatigue measurement and is validated among patients with cancer [24]. In addition, CONUT is an objective bio-marker of nutritional status that is calculated from common clinical laboratory measures, making it practical to use in a clinical setting. Importantly, this study is one of the first to specifically explore the relationships between nutritional status, omega-3 uptake from fish oil supplements, and cancer-related fatigue, lending insight into why different fish oil supplement studies arrive at disparate findings.

## 5. Conclusions

Better nutritional status was associated with a greater increase in circulating omega-3 fatty acids with fish oil supplementation, and greater increases in circulating omega-3 fatty acids were associated with greater improvements in fatigue. These results should be tested for replication in a follow-up study. More broadly, strategies to improve nutrient uptake should be further explored to address malnutrition, cancer-related fatigue, and other issues of supportive care among cancer survivors.

## Figures and Tables

**Figure 1 nutrients-14-00184-f001:**
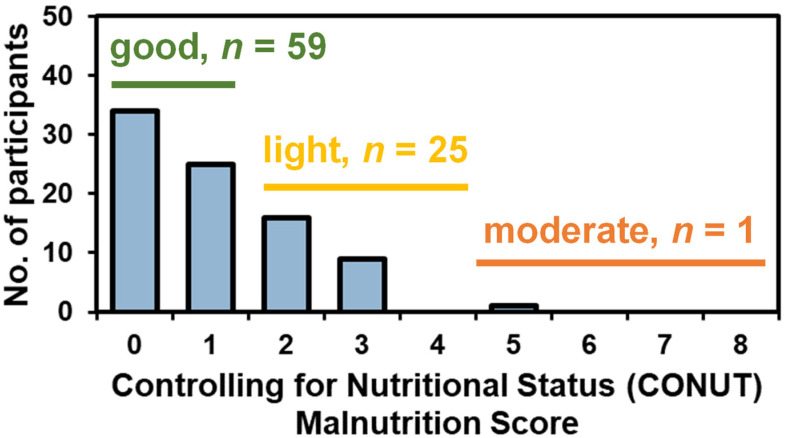
Controlling for nutritional status (CONUT) distribution in our cohort (*n* = 85).

**Figure 2 nutrients-14-00184-f002:**
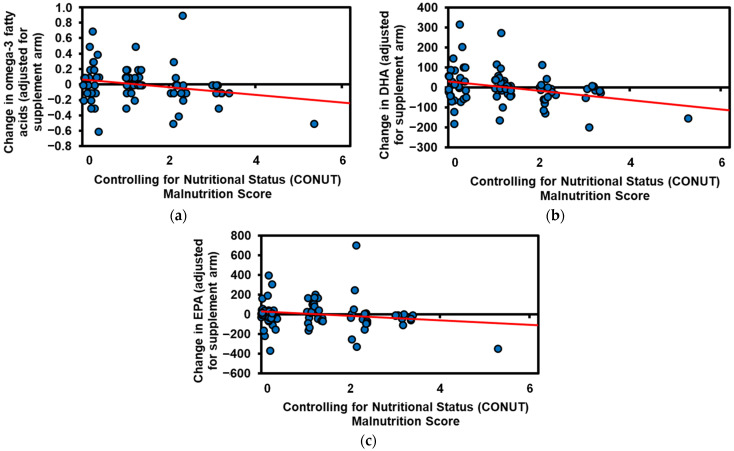
Associations between baseline Controlling for Nutritional Status (CONUT) score and change in serum (**a**) total omega-3, (**b**) docosahexaenoic acid (DHA), and (**c**) eicosapentaenoic acid (EPA) concentrations. A higher CONUT score indicates more severe malnutrition. The red line depicts the line of best fit.

**Table 1 nutrients-14-00184-t001:** Demographics and clinical characteristics (*n* = 85) at baseline.

	Good Nutritional Status (*n* = 59)	Light-Moderate Malnutrition (*n* = 26)	*p*-Value *
	Mean ± SD or *n* (Percent)	Mean ± SD or *n* (Percent)	
**Age (years)**	61.3 ± 9.4	61.2 ± 10.7	0.97
**Body mass index (kg/m^2^)**	31.5 ± 6.0	32.8 ± 8.3	0.47
**Race (self-identified)**			0.28
White	56 (94.9%)	23 (88.5%)	
Other	3 (5.1%)	3 (11.5%)	
**Menopausal status**			0.88
Pre-menopausal	4 (6.8%)	2 (7.7%)	
Peri- or post-menopausal or medically induced	55 (93.2%)	24 (92.3%)	
**Marital status**			0.29
Married or long-term relationship	41 (69.5%)	15 (57.7%)	
Divorced, separated, single, or widowed	16 (27.1%)	10 (38.5%)	
**Education**			0.44
Up to a high school degree	20 (33.9%)	11 (42.3%)	
At least some college	37 (62.7%)	14 (53.8%)	
**Cancer stage**			0.27
0	4 (6.8%)	0 (0%)	
1	23 (39.0%)	13 (50.0%)	
2	26 (44.1%)	8 (30.8%)	
3	5 (8.5%)	4 (15.4%)	
**Cancer treatment**			
Surgery (yes)	56 (94.9%)	26 (100%)	0.24
Months since treatment	17.7 (9.7%)	19.0 (9.4%)	0.57
Chemotherapy (yes)	30 (50.8%)	11 (42.3%)	0.47
Months since treatment	17.0 (8.3%)	17.8 (9.2%)	0.79
Radiation therapy (yes)	41 (69.5%)	22 (84.6%)	0.14
Months since treatment	19.4 (28.8%)	17.4 (9.1%)	0.67
**Current hormonal therapy**			0.045
No	19 (32.2%)	3 (11.5%)	
Yes	40 (67.8%)	23 (88.5%)	
**Fatty acid concentration**			
Total omega-3 fatty acids (mM)	0.33 ± 0.10	0.27 ± 0.12	0.013
Docosahexaenoic acid (DHA, μM)	135.24 ± 47.70	125.77 ± 56.08	0.458
Eicosapentaenoic acid (EPA, μM)	70.68 ± 34.08	48.12 ± 35.23	0.009
α-Linolenic acid (ALA, μM)	87.61 ± 37.70	70.08 ± 28.77	0.022
Total omega-6 fatty acids (mM)	4.65 ± 0.79	3.92 ± 0.77	<0.001
Linoleic acid (μM)	3497.3 ± 562.4	2958.1 ± 546.0	<0.001
Arachidonic acid (μM)	844.95 ± 243.42	721.38 ± 240.89	0.035
Omega-6:omega-3 ratio	14.88 ± 4.06	16.56 ± 6.12	0.207

* *p*-value derived from comparing those with good vs. light-moderate malnutrition via a two-sided *t*-test for continuous variables or Pearson chi-square test for categorical variables.

**Table 2 nutrients-14-00184-t002:** Change in serum fatty acid concentrations with 6 weeks of fish oil, soybean oil, or fish+soybean oil supplementation (*n* = 80). *p*-values are derived from analysis of variance comparing changes in fatty acids between groups.

	6 g Fish Oil (*n* = 30)	3 g Fish Oil + 3 g Soybean Oil (*n* = 23)	6 g Soybean Oil (*n* = 30)	*p*-Value
Fatty Acid	Mean ± SE	Mean ± SE	Mean ± SE	
Total omega-3 fatty acids (mM)	0.59 ± 0.33	0.38 ± 0.19	−0.01 ± 0.09	<0.001
Docosahexaenoic acid (DHA, µM)	207.13 ± 101.79	164.83 ± 80.48	−8.30 ± 44.82	<0.001
Eicosapentaenoic acid (EPA, µM)	358.00 ± 223.02	187.78 ± 107.86	−5.04 ± 31.44	<0.001
Total omega-6 fatty acids (mM)	−0.56 ± 0.67	−0.11 ± 0.73	0.05 ± 0.83	0.008
Linoleic acid (µM)	−257.43 ± 105.77	54.39 ± 120.80	85.26 ± 111.49	0.054
Arachidonic acid (µM)	−208.20 ± 172.49	−105.48 ± 169.55	−31.78 ± 180.74	0.001
Omega-6:omega-3 ratio	−11.19 ± 6.94	−9.13 ± 3.80	0.57 ± 2.94	<0.001

**Table 3 nutrients-14-00184-t003:** Changes in serum fatty acid concentrations as a function of baseline nutritional status (*n* = 80). Models are adjusted for supplement group.

Dependent Variable	Effect Estimate	Standard Error	*p*-Value
Total omega-3 fatty acids (mM)	−0.0492	0.017	0.005
Docosahexaenoic acid (DHA, μM)	−20.233	6.764	0.003
Eicosapentaenoic acid (EPA, μM)	−19.400	9.055	0.032
Total omega-6 fatty acids (mM)	0.026	0.069	0.705
Linoleic acid (µM)	6.160	60.782	0.919
Arachidonic acid (µM)	15.309	15.489	0.323
Omega-6:omega-3 ratio	0.596	0.470	0.205

**Table 4 nutrients-14-00184-t004:** Associations between change in circulating omega-3 fatty acids and change in fatigue among cancer survivors who were supplemented with fish oil (*n* = 53).

Dependent Variable	Estimate (β) *^,^^†^	Std Error	*p*-Value
Multidimensional Fatigue Symptom Inventory-Short Form (MFSI) total score	−11.731	6.500	0.071
General fatigue	−2.745	2.347	0.242
Physical fatigue	−4.532	1.983	0.022
Emotional fatigue	0.257	1.484	0.863
Mental fatigue	−1.051	1.321	0.426
Vigor	4.852	2.041	0.018

* A negative β indicates that a greater change in omega-3′s was associated with a greater reduction in fatigue for general, physical, emotional, and mental fatigue and the total score. For vigor, a positive β indicates that a greater change in omega-3′s was associated with a greater increase in vigor. ^†^ Models are adjusted for age and body mass index.

## Data Availability

Raw data and statistical coding is available from A.S.K. or L.J.P. upon reasonable request.

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
