# Peer review of "Nutritional Status Predicts Fatty Acid Uptake from Fish and Soybean Oil Supplements for Treatment of Cancer-Related Fatigue: Results from a Phase II Nationwide Study"

_nutrients, 2021, doi:10.3390/nu14010184_

Round 1
Reviewer 1 Report
Well written article on fish oil supplements correlated with scoring of fatigue. This is a secondary analysis of a randomized three arm study. I have no major concerns on the methodology Some concerns about discussion.
1. Additional weakness of this study should be discussed. The use of three laboratory parameters to judge malnutrition is a minimal assessment . Body composition needs to be mentioned as sarcopenic obesity is likely given the raised BMI in the majority of patients. Eg Does severe sarcopenia correlate with fatigue
2. While briefly discussed that one micronutrient should not isolated in nutritional assessment and intervention there is no additional discussion on potential other micronutrients that should be evaluated in future studies.
3. Is fatigue more prevalent in the chemotherapy patients?
4. The fatigue assessment is a validated tool but very subjective. Mention other tools of assessing fatigue
Reviewer 2 Report
The manuscript by Kleckner et al. investigates the benefits of omega-3 supplementation in cancer-related fatigue and examines if the outcomes related to the omega-3 supplementation is affected by nutritional status. To test their hypothesis, they used results from their previously published large fish oil supplementation trial among breast cancer survivors (JNCI Cancer Spectrum, 2019). Fatigue was assessed using MFSI (Multidimensional Fatigue Symptom Inventory-Short Form) and nutritional status was evaluated using a CONUT (controlling nutritional status tool) parameter that considers total lymphocyte counts, albumin, and cholesterol measurements. They report that greater improvement of fatigue is associated with higher serum levels of omega-3 fatty acids, and that baseline nutritional status would affect omega-3 uptake. The manuscript is well-written and it has a clear hypothesis. While this study tests an interesting and important hypothesis, some clarifications are needed.
- Do the patients in the trial take any medications? Since they are overweight or obese, do they have diabetes or any other comorbidities that might contribute to the fatigue?
- Why do authors think soybean oil was more effective than omega-3 in reducing fatigue? Since this is the case please also show change in omega-6 fatty acids like ARA or LA as a function of CONUT score.
- Table 2 – it is not clear what statistics was performed to generate the p values here. Please clarify. There are 3 groups on the table. Was a multiple comparisons test performed? Also, what do these values indicate, difference from baseline? What do the negative values denote?
- Table 3 needs some clarification. It is not clear why the values are negative. I am assuming these are differences from baseline. For readers who are not in the field of Epidemiology, it would be best if the authors would clarify the ‘effect estimate’ on Table 3.
- Figure 2- please clarify what the red line indicates.
- Line 170- Authors should clarify how they came up with 0.36 mM increase with 6 weeks of supplementation. Also, the beginning of this sentence has a grammatical error.
“Our models estimates that, breast cancer patients with good nutrition status (CONUT=0) vs. those with malnutrition (CONUT=3) who supplement with 3 g of fish oil daily for 6 weeks would see an increase of 0.36 mM vs. only 0.21 mg/mL circulating omega-3 fatty acids—a 70% greater increase. Extrapolating these models, a breast cancer survivor with a CONUT score of 3 would theoretically have to take 26% more fish oil to get the same 0.3-mM increase in circulating omega-3’s than a survivor with good nutrition status (a CONUT score of 0).”
7. Line 265- Authors mention a study that examined omega-3 fatty acid intake in 633 breast cancer survivors (below). This contradicts with what they mention at the beginning of the Discussion that their study is the largest omega-3 supplementation study in breast cancer survivors. Table 1 indicates n=85 in this study. Please clarify.
"Similarly, in a cross-sectional study among 633 breast cancer survivors, higher intake of omega-3 fatty acids was associated with a lower odds of fatigue...."
